# The direct and indirect effects of community beliefs and attitudes on postpartum contraceptive method choice among young women ages 15–24 in Nigeria

David K. Guilkey[1,2ᴼ], Ilene S. Speizer[1,3ᴼ]*

1 Carolina Population Center, University of North Carolina at Chapel Hill, Chapel Hill, NC, United States of America, 2 Department of Economics, University of North Carolina at Chapel Hill, Chapel Hill, NC, United States of America, 3 Department of Maternal and Child Health, Gillings School of Global Public Health, University of North Carolina at Chapel Hill, Chapel Hill, NC, United States of America

ᴼ These authors contributed equally to this work.
* speizer@email.unc.edu

**Data Availability Statement:** The data for this study come from the MLE Nigeria Sustainability Study and are publicly available from this site: https://dataverse.unc.edu/dataverse/mle_nigeria.

## Abstract

Understanding what factors influence postpartum contraceptive use among young people (ages 15–24 years) is important since this group often has closely spaced and unintended births. Using secondary data gathered for an evaluation of a Bill & Melinda Gates Foundation funded initiative designed to increase modern contraceptive use in select urban areas of Nigeria, we determine the direct and indirect effects of community beliefs and attitudes on adolescent and youth postpartum contraceptive method choice. Our statistical methods control for the endogenous timing of the initiation of sexual activity and the timing and number of births to each respondent by simultaneous estimation of equations for these choices with the choice of postpartum contraceptive method. We find that community beliefs and attitudes have important effects on our primary outcome of postpartum contraceptive use and we quantify the size of both direct and indirect effects on postpartum contraceptive method choice using simulations. The findings from this study can be used to inform programs seeking to increase young women's postpartum contraceptive use for healthy spacing and timing of births.

## Introduction

Pregnancies that are too closely spaced pose risks for the mother as well as for the newborn [1–5]. The World Health Organization (WHO) recommends three-year birth intervals; that is, following a pregnancy that women wait at least 24 months before trying to get pregnant again [6]. Despite these recommendations, many women in poor countries are at risk of unintended and closely spaced births. A key strategy to support lengthening birth intervals is provision of postpartum contraceptive services, particularly in the immediate postpartum period [7]. Recent estimates demonstrate that about 61% of women are not using an effective method of contraception within 24 months postpartum, increasing their risk of short birth intervals [8].

**Funding:** This work was supported, in whole or in part, by the Bill & Melinda Gates Foundation [INV-009814]. Under the grant conditions of the Foundation, a Creative Commons Attribution 4.0 Generic License has already been assigned to the Author Accepted Manuscript version that might arise from this submission. We also received general support from the Population Research Infrastructure Program through an award to the Carolina Population Center (P2C HD050924) at The University of North Carolina at Chapel Hill. The contents of this article are solely the responsibility of the authors and do not necessarily represent the official views of CPC or the Bill & Melinda Gates Foundation.

**Competing interests:** The authors have declared that no competing interests exist.

A group that is often overlooked by postpartum contraception programs are young mothers under the age of 25 [9]. Young women who have early and closely spaced births are at increased risk for eclampsia, puerperal endometritis, systemic infections, low birth weight, preterm delivery, and severe neonatal conditions [10]. Further, young women who have early and repeat pregnancies also risk dropping out of school, are less likely to participate in the labor force, earn less income, and live in poorer households [11, 12]. Young mothers are often ignored by postpartum family planning (FP) programs because of societal pressures for them to marry and demonstrate their fecundability [13]. While some births among young women are planned, many are unplanned or unwanted [14, 15]. Young women have numerous hurdles to using contraception to avoid an unplanned pregnancy. An important hurdle is provider bias that results in providers refusing services to young women because of societal norms about fertility or sexuality [16]. Further, young women also often lack agency, ability, or decision-making power to seek and use contraceptive services [17]. In a recent review of interventions to prevent unintended pregnancy and short birth intervals among adolescents, Norton and colleagues [9] find that effective programs fall into five general categories. These were: a) offering contraceptive services and information that included outreach and support; b) immediate postpartum counseling and services following delivery; c) increasing adolescent's life planning skills, including contraception as part of life planning; d) social and behavior change programming that identify how contraception can help positive youth development; and e) offering mentoring and goal setting for young people. The authors note the importance of cultural norms on affecting program strategies and young people's ability to act on their postpartum contraceptive intentions and decisions [9].

This paper contributes to the literature on strategies to strengthen postpartum programming for young people by examining the direct and indirect effects of community beliefs and attitudes on adolescent and youth ages 15–24 postpartum contraceptive method choice. The analysis is guided by previous work that has studied social norms, peer effects and diffusion on contraceptive use and unintended pregnancy prevention among women of all ages [18–21]. Further, numerous social and gender norms, attitudes, and beliefs have been identified that influence young people's sexual and reproductive health behaviors, including adoption of contraception to prevent an unintended pregnancy [17, 22]. These include cultural norms that affect both provider attitudes about provision of methods to young women as well as those norms that affect young people's willingness to visit a health facility [17]. Further, there are social pressures to conceive and demonstrate fecundability, as well as the expectation that contraception is only for people who are married or in union [17]. Finally, typically young people (and people of all ages) have perceptions about contraceptive use that may be based on myths or misconceptions; these have been found to be important barriers to use among women, including young women, in numerous settings [17, 23, 24]. To date, few studies have examined how norms, attitudes, and beliefs influence young women's postpartum contraceptive behaviors; this is the contribution of this paper that uses data from women 15–24 years of age in Nigeria to examine how norms, attitudes, and beliefs are related to contraceptive decision-making in the one-year postpartum period.

## Study context

Nigeria is the most populous country in Africa with about half of the population living in urban areas. The total fertility rate (TFR) in Nigeria in 2018 was 5.3 births per woman; the TFR is higher in rural areas (5.9) compared to urban areas (4.5) [25]. The population is young with 46% of the population under age 15 with an additional 15.6% of the population aged 15–24 [25]. The median age of childbearing in Nigeria was 20.4 years in 2018 with urban women

starting childbearing later (22.3 years) than rural women (19.0 years) [25]. The average birth interval among women aged 15–19 was 26.9 months and among women 20–29 it was 28.8 months [25]. This is substantially below the recommended 36-month birth interval [6].

This study includes data from women in three cities: Kaduna in Kaduna state in the North West region and Jos (Plateau state) and Ilorin (Kwara state) in the North Central region. These states differ in fertility rates with Kaduna state having the highest TFR (5.9) followed by Kwara state (5.2) and then Plateau state (4.7). Median age at first birth follows a similar pattern with Kaduna state having the lowest age (18.9 years) followed by Plateau (20.6 years) and Kwara state (21.4 years) [25]. The cities are similar size with populations near one million. Kaduna and Jos are predominately Hausa speaking while Ilorin is predominately Yoruba speaking. In Ilorin and Kaduna, about two-thirds of the population is Muslim whereas in Jos the percentage is around two-fifths.

## Methods

### Data set

The Nigerian Urban Reproductive Health Initiative (NURHI), funded in 2009 by the Bill & Melinda Gates Foundation (BMGF), aimed to increase FP access and modern contraceptive use in select urban areas. (Details of the NURHI program can be found elsewhere [26]). Phase I (2009 to early 2015) of the NURHI project was implemented in six cities: Abuja, Benin City, Ibadan, Ilorin, Kaduna, and Zaria. In 2015, Phase 2 of the NURHI project was launched (Phase 2 was 2015–2020). In Phase 2, activities continued in Oyo (incorporating Ibadan) and Kaduna state (incorporating Kaduna city and Zaria) and expanded to Lagos. Program activities ended in the other cities.

The Measurement, Learning & Evaluation (MLE) project (also funded by BMGF) at the Carolina Population Center at the University of North Carolina at Chapel Hill conducted the evaluation of the NURHI Phase I program [27]. The main data collection periods for the NURHI Phase I evaluation were baseline (2010/2011); midterm (2012); and endline data collection (2014/2015). In 2017, another follow-up round of cross-sectional representative survey data were collected in Kaduna, Ilorin, and Jos, the capital city of Plateau state. This survey was part of the NURHI Sustainability Study led by the MLE team and was meant to determine if program activities continued after the donor left. The sampling design varied by site but in each site, data were collected from a representative sample of women ages 15–49. In Kaduna and Ilorin, using the study clusters sampled in the endline representative survey in 2015 (n = 45 clusters in Kaduna and 54 in Ilorin), the same clusters were visited and all households in those clusters were included in the survey (this permitted having a longitudinal sample as well as a cross-sectional representative sample of women). For Jos, where the NURHI evaluation did not take place, a representative sample of 56 clusters were selected based on the 2006 Nigeria census and in each cluster, 33 households were randomly chosen. All women of reproductive age (15–49 years) living in the selected households were eligible to participate in the survey following provision of verbal informed consent; interviewers signed the consent form to indicate that the consent forms were read to the participants, and participants verbally agreed to participate. All consent procedures were approved by the Institutional Review Board at the University of North Carolina at Chapel Hill and by the National Health Research Ethics Committee of Nigeria (NHREC) in Nigeria. A total of 4,673 women from Kaduna, 2,209 women from Ilorin, and 2,003 women from Jos were surveyed. This study focuses on young women aged 15–24: 1,941 women from Kaduna, 822 from Ilorin, and 790 from Jos. Weights are used to make the samples representative at the city level. Demographic characteristics of the study sample are presented in Table 1.

## Variables

**Dependent variables.** The data for this analysis of postpartum contraceptive method use come from a retrospective reproductive calendar for the five years before the survey. The overall survey tool that included the reproductive calendar was based on the Measurement, Learning & Evaluation project validated questionnaires that were previously used in Nigeria as part of the longitudinal evaluation. (See data availability statement for information on accessing the questionnaires and data from the Measurement, Learning & Evaluation project and from this follow up NURHI Sustainability Study.) The main outcome of this study is use of a contraceptive method in the first year following the birth of each child. A one-year postpartum period has been selected as the follow-up time in this analysis as this is a key time to target women who are visiting a health facility for postpartum and well-baby visits and evidence suggests that a majority of women in this period are not using an effective method of contraception to avoid an unintended pregnancy [28]. A longer period (i.e., up to 24 months postpartum) could be examined; however, with the data available, this would mean significant right censoring in the data which we prefer to avoid for this analysis of young women.

Given that not all women in the sample experienced a pregnancy or ever had sex by the time of the survey, we model the selection into a postpartum state by also modeling the timing of first sex and the timing of births. Thus, while the main focus of this study is postpartum contraceptive use, we also model two additional reproductive health events (experience of first sex and experience of a birth) as part of a multi-equation analysis approach. In total, there were 1,082 births by age 24 and most were first or second births. For ten of these births, key information was missing, and these births were dropped. The remaining 1,072 births allow us to examine 1,072, possible episodes of postpartum contraceptive use in the year following each of these births. Notably, some women contributed multiple observations (i.e., one-year event periods) if they had multiple births in the five-year calendar period. Among the women who adopted a method in the one-year postpartum period, we categorize the method used as a traditional method (rhythm, withdrawal, or another natural method, including the one woman who reported Standard Days Method), lactational amenorrhea, a method available at shops or pharmacies (pills, emergency contraception, condoms), or a facility-based method (injectable, IUD, implants). Notably, most of the use in the latter category is injectable contraceptive use which might be promoted during a postpartum maternity or well-baby visit. Any woman who reported two methods in the same month was coded as using the more effective of those methods in that month. When a woman reported multiple methods in the one-year postpartum period, she was coded based on the most effective method she used in that period. Therefore, a woman who reported first using LAM and then switching to injectable contraceptives would be coded as an injectable method user in the one-year postpartum period. Further, for LAM users, this was only included in the calendar when it was reported by the woman as the method she was using to avoid a pregnancy as opposed to classifying women as LAM users based on relevant behaviors (i.e., timing postpartum, whether she has resumed sexual activity, and if she is fully breastfeeding).

Also shown in Table 1 is the timing of marriage for the respondents. We had originally planned to include this as an additional dependent variable; however, it was too collinear with the timing of first sex and was dropped. We chose to use first sex instead of first marriage as first sex may precede marriage and puts the women at risk of a birth which is the focus of this analysis that examines postpartum contraceptive use. The average age at first sex is about half a year earlier than the average age of first marriage.

**Community beliefs and attitude variables.** The survey asked numerous questions to gauge attitudes and beliefs about use of contraceptives that may be associated with the choice

**Table 1. Descriptive characteristics of the sample of women ages 15–24 at the time of the survey, Ilorin, Kaduna, and Jos, Nigeria, 2017.**

| Individual Level Characteristics | | Reproductive Variables | |
|---|---|---|---|
| Age (average in years) | 19.09 | Ever had sex (%) | 38.11 |
| Education level (average in years) | 9.23 | | |
| City (%) | | Average age at first sex among those who ever had sex (years) | 17.71 |
| Ilorin | 33.82 | | |
| Kaduna | 42.02 | Ever married (%) | 1.10 |
| Jos | 24.16 | | |
| Religion (%) | | Average age at first marriage among those ever married (years) | 18.27 |
| Catholic | 5.89 | | |
| Protestant | 32.47 | | |
| Muslim | 61.63 | | |
| Other/missing | 0.02 | | |
| | | Ever given birth (%) | 18.89 |
| Religiosity (%) | 77.53 | Average age at first birth among those who have given birth (years) | 18.81 |
| Strongly religious | 22.18 | | |
| Somewhat religious/not religious | | | |
| | | Birth order of included birth | |
| Duration of residence (%) | | First birth (%) | 58.79 |
| Lived in city ≤ 1 year | 8.79 | Second birth (%) | 27.22 |
| Lived in city 1–5 years | 11.70 | Higher order birth (%) | 13.99 |
| Lived in city more than 5 years | 79.51 | | |
| | | **Contraceptive use in year postpartum (%)** | (n = 1072)births |
| | | Non-user | 61.14 |
| | | Used traditional method | 12.70 |
| | | Lactational amenorrhea method (LAM) | 4.46 |
| | | Shop/pharmacy-based method[a] | 10.71 |
| | | *Daily pill* | *3.60* |
| | | *Emergency contraception* | *1.67* |
| | | *Male condom* | *5.44* |
| | | Facility-based method[b] | 11.00 |
| | | *Injectable* | *7.24* |
| | | *Implant* | *3.44* |
| | | *IUD* | *0.32* |

All values shown are weighted; unweighted number of observations is 3,454. Final models do not include age at marriage because of high correlation with timing of first sex.

[a]Shop/pharmacy-based methods include: male condoms and daily and emergency. Facility-based methods include: IUD, implant, injectables.

to use and method choice. A smaller number of attitudes and beliefs are included that could be hypothesized to affect the timing of first sex and the timing and number of births. In Table 2, we provide the wording for the questions grouped by the three outcomes that they may correspond to. For contraceptive use, we further break down the responses into three categories: contraceptive norms, provider norms, and myth norms. For the timing of first sex, we have one question on the belief about the level of sexual activity among unmarried girls and we have three questions that are hypothesized to affect the timing of births.

Because of the large number of variables, we used principal components analysis to construct indexes for contraceptive norms, provider norms, myth norms, and childbearing norms using all the variables listed in these categories in Table 2. In all cases, we used the scoring

**Table 2. Community beliefs and attitudes variables among all female respondents ages 15–49 in three cities in Nigeria.**

| Attitude/Belief | Question | Weighted Average for All Respondents |
|---|---|---|
| **Variables hypothesized to affect contraceptive method choice** | | |
| Contraceptive norm | • Best friend would approve of family planning (% yes)<br>• Do you think there are some people within this community who will praise, encourage, or talk favorably about you if they knew that you were using a family planning or a contraceptive method? (% yes)<br>• Among unmarried girls who are sexually active in your community, how many do you think are using contraception: none, some, most, or all? (% all or most) (don't know coded as zero)<br>• Among your close friends, how many do you think are using contraception: none, some, most, or all? (% all or most) (don't know coded as zero)<br>• In the past year, have you heard or seen a local government official speaking publicly **in favor of** family planning/childbirth spacing? (% yes)<br>• In the past year, have you heard or seen a religious leader speaking publicly **in favor of** family planning/childbirth spacing? (% yes) | 53.0%<br>44.9%<br>13.3%<br>19.1%<br>52.8%<br>46.4% |
| Provider norm | • Family planning providers around here treat clients very badly. (% strongly agree/agree)<br>• Women don't like the way they are treated in FP clinics around here (% strongly agree/agree)<br>• FP sellers/providers make women like you feel bad when obtaining contraceptives (% strongly agree/agree) | 9.5%<br>10.0%<br>10.0% |
| Myth norm | • People who use family planning end up with health problems. (% strongly agree/agree)<br>• Use of a contraceptive injection can make a woman permanently infertile (% strongly agree/agree)<br>• Contraceptives can harm your womb (% strongly agree/agree)<br>• Contraceptives reduce women's sexual urge (% strongly agree/agree)<br>• Contraceptives can cause cancer (% strongly agree/agree)<br>• Contraceptives can give you deformed babies (% strongly agree/agree)<br>• Contraceptives are dangerous to your health (% strongly agree/agree)<br>• Women who use family planning /child birth spacing may become promiscuous (%<br>• strongly agree/agree) | 38.3%<br>27.8%<br>29.0%<br>15.4%<br>19.8%<br>16.3%<br>36.6%<br>16.7% |
| **Variable hypothesized to affect timing of first sex** | | |
| Sexual activity among unmarried girls (all or most) | • How many unmarried girls in your community do you think are sexually active: none, some, most, all? (% all or most) | 25.1% |
| **Variable hypothesized to affect timing of births** | | |
| Childbearing norms | • Strongly agree or agree that it is good to have many children because one is not sure who among them will survive to care for the parents at old age. (% strongly agree or agree)<br>• Ideal age to get pregnant (or have a child) for the first time (years) coded 1 if under 20, zero if older<br>• Ideal number of months between two children (months) *coded 1 if less than 2 years, zero if longer* | 36.1%<br>17.6%<br>2.6% |

Note: Provider norms not included in final model because collinear with the myth norm and the general contraceptive norm. Beliefs and attitudes calculated for full sample of women (n = 8,885), however, some of the samples are smaller due to a small number of missing responses to the relevant questions to calculate the norms.

coefficients from the first principal component to construct the indexes. In all cases except for contraceptive norms, the first principal component explained over 50% of the variation in the variables while for contraceptive norms, the first component explained 32% of the variation but this was nearly twice as much as the second component.

We then used these four indexes and the single variable hypothesized to affect timing of first sex to calculate proxies for community beliefs and attitudes. The community level variables were calculated using the survey primary sampling unit (PSU) as a proxy for community as is common in the literature [29, 30] and averaging the individual level responses to the questions. In calculating the averages, we drop the index woman from her average to remove individual level bias. We do not restrict the sample to women ages 15 to 24 years when calculating the averages but rather use all 8,885 respondents from the three cities in the survey. We did this because young women could be influenced by community members of all ages and the larger sample size helps to reduce the effects of measurement error by having a substantial

number of observations for each PSU. We have a total of 155 PSU with an average of 57 respondents per PSU.

**Individual level explanatory variables.** Control variables are shown on the left side of Table 1 and were selected based on prior studies that have examined fertility and family planning behaviors among adolescents and youth [31, 32]. These include a continuous age variable, a continuous education variable, city of residence (Ilorin, Kaduna, or Jos), religion (Muslim, Catholic, Protestant, other/missing), religiosity (strongly religious or somewhat religious vs. not religious), and duration of residence (lived in city $\leq 1$ year; lived in city 1–5 years; and lived in city more than 5 years). Note that Kaduna had been a NURHI program city during Phase 1 and Phase 2 while Ilorin was included until 2015 and Jos was never in the program.

## Empirical model

Our empirical model consists of three equations. For our primary outcome, postpartum contraceptive method choice in the year after the birth of each child, we specify a multinomial logit model. We model the timing of first sex as a discrete time hazard model and the timing of births as a multiple spell discrete-time hazard model. Clearly, these are a set of sequential outcomes with sex preceding births and births preceding postpartum contraceptive use. However, if there are unobservable variables that affect the three decisions, then selection bias is introduced if one does not account for it. For example, the respondent may have knowledge of her fecundity which could affect the timing of first sex, the timing of births and postpartum method choice. The seminal work on this type of selection bias caused by common unobservables was undertaken in the 1970s by Heckman [33]. Selection bias in hazard models due to unobservables, such as our equations for the timing of first sex and timing of births, are especially problematic as demonstrated by Heckman and Singer [34]. Methods used to correct for selection bias, similar to the ones we use in this paper, have frequently been applied in fertility related research. This includes, for example, a study on fertility and female education in Indonesia [35]; a study of the effects of the availability of child care on fertility in Norway [36]; and a study of marriage duration and fertility timing in the U.S. [37].

**Statistical specification.** Postpartum contraceptive use in the year following a birth is our primary outcome. However, we write out the statistical model in the sequential order in which decisions are made starting with a discrete time hazard equation for the timing of first sex:

$$\ln[\frac{P(S_{ijt} = 1|S_{ij,t-1} = 0)}{P(S_{ijt} = 0|S_{ij,t-1} = 0)}] = X_{ijt}^S \beta^S + C_{ij}^S \alpha^S + \mu_j^S + \varepsilon_{ij}^S. \qquad (1)$$

Where the dependent variable is the log odds that woman i (i = 1,2,. . .,$N_j$) from community j (j = 1,2,. . .,M) had first sex at time t (t = 1,2,. . .,$T_{ij}$) conditional on not having had first sex up to time t. Only four women reported an age at first sex younger than 12 and we set their age at first sex to 12. We follow each woman yearly from age 12 until right censoring occurs at her age at the time of the survey or at age 24. By starting the process at age 12, we eliminate the possibility of left censoring which introduces additional complications into hazards models. We can do this because only four respondents initiated sex prior to age 12 and we simply set age at first sex to 12 for these four individuals. Note that time is not distinguishable from age in our data (age = t+11) and so by including age and age squared in the model we are also controlling for duration dependence. The X's represent a set of exogenous factors that affect age at first sex and include time varying age and age squared. Education is also time varying where we use the respondent's terminal education at the time of the survey and assume a yearly education progression starting at age six until she reaches her terminal level. The other independent variables in "X" (e.g., city, religion, duration of residence) are time invariant. Finally, $C_{ij}^S$

represents the community variable that is hypothesized to affect the onset of sexual activity. Since we drop the index woman from the average associated with her, we have an "i" subscript on C.

We include both a time invariant community level error ($\mu_j^S$) and a time invariant individual level error ($\varepsilon_{ij}^S$). The presence of the community level error allows observations for respondents in the same community to be correlated due to community level unobservables while the time invariant individual level error allows for correlation due to individual level unobservables. Implicit in the logit specification is a time varying individual level error that is assumed to follow the logistic cumulative distribution.

The second equation is a multiple spell discrete time hazard model for the timing of births starting from the age the respondent reported first having sex until censoring occurs at the woman's age at the time of the survey or age 24:

$$\ln\left[\frac{P(B_{ijtk}=1|B_{ij,t-1,k}=0)}{P(B_{ijtk}=0|B_{ij,t-1,k}=0)}\right] = X_{ijt}^B \beta^B + C_{ij}^B \alpha^B + \mu_j^B + \varepsilon_{ij}^B. \tag{2}$$

Where the dependent variable is the log odds that woman i from community j had birth k (k = 1,2,. . .,6) at time t given that birth k had not occurred prior to time t. The statistical specification of the right-hand-side of Eq (2) is the same as Eq (1) and there is overlap in the X's. However, there are differences that are detailed below. In addition, this equation includes the endogenous age at first sex. We model all births to a respondent up to age 24 and we restrict the coefficients to be the same across birth intervals since there are 1,072 total births and 57% are first births and 29% are second births. Notably, two women in our sample of young women reported six births.

The third equation is for our primary outcome, postpartum contraceptive method choice:

$$\ln\left[\frac{P(M_{ijt}=m)}{P(M_{ijt}=1)}\right] = X_{ijt}^M \beta_m^M + C_{ij}^M \alpha_m^M + \mu_{mj}^M + \varepsilon_{mij}^M. \tag{3}$$

Where the dependent variable is the log odds that woman i from community j at time t used method m (m = 2,3,4,5) in the year following each birth relative to not using any method ($M_{ijt}$ = 1). As with Eqs (1) and (2) there is overlap in the explanatory variables but some of the variables are unique to this equation and include indicators for the birth order of the child. Postpartum contraceptive use is coded as follows: 1: nonuse; 2: traditional method use; 3: lactational amenorrhea method (LAM); 4: shop/pharmacy-based methods; and 5: facility-based methods; definitions of shop/pharmacy-based and facility-based methods are given above.

The $\mu$'s and $\varepsilon$'s serve multiple purposes in our full information joint estimation strategy. First, in the timing of first sex equation, their inclusion along with our maximum likelihood estimation strategy controls for the selection on unobservables into a later age for the initiation of sexual activity. Methods that do not allow for this type of unobserved heterogeneity can lead to biased results [38, 39]. The same issues are present in the birth equation where we must control for selection into later ages for births and the selection into higher order births. Finally, by allowing the $\mu$'s and $\varepsilon$'s to be correlated across the three equations, we control for the selection into the births equation following the initiation of sexual activity and the inclusion of the endogenous variable age at first sex in both the birth and method choice equations and the inclusion of the birth order indicators in the method choice equation. We use a variation of the discrete factor approximation method (DFAM) to estimate the distribution of the $\mu$'s and $\varepsilon$'s along with the model's other parameters specified in Eqs (1)–(3). The Fortran 77 program we use (referred to as LEO) was written and copyrighted by Jeffrey Rous from the Department

of Economics at North Texas State University. The manual for the package describes the program, specifies the likelihood function, and provides detailed instructions on how to write a setup file that the program uses to specify the set of equations and read in the raw data as an ascii file. A copy of the manual for LEO and an example of the setup file are in the S2 File. Leo maximizes the likelihood function using the Goldfeld-Quandt non-linear optimization package (www.quandt.com/gqopt.html) that is marketed by Richard Quandt from Princeton University. The package includes a large number of non-linear optimization packages. The DFAM (LEO) program can invoke two of the packages: the Davidon-Fletcher-Powell (DFP) algorithm and the quadratic hill climbing algorithm (GRADX). We used DFP as a first step and then switched to GRADX when we got close to the maximum of the likelihood function to get more precision and a better estimate of the covariance matrix of the parameter estimates. In both cases, we used the default versions of the algorithms. The method is explained more fully in the S1 File along with a table that presents the estimation results for these ancillary parameters.

Note that several of the regressors are time varying and this is the main source of identification for the model [40]. Although this highly non-linear model is technically identified without exclusion restrictions [41].

A final issue that needs to be addressed is related to the use of community averages to create measures for community beliefs and attitudes. The approach used in this study to control for the potential endogeneity of these community variables is to drop the index woman from the calculation of the average. A similar strategy is used in the peer effects literature [42–45]. However, a key difference is that in much of the peer effect literature the averages (minus the index respondent) that are used as explanatory variables are averaged values of the dependent variable for the index respondent's peers. For example, average sexual activity among peers affecting the respondent's sexual activity [42] or average achievement of peers affecting an individual's level of achievement [43]. This leads to the well-known social reflection problem [44] where self-selection of peers causes the average to be endogenous. The typical solution is to use instrumental variables methods to correct for bias but these methods frequently suffer from weak instruments which make the results unreliable [45].

Our method does several things to mitigate bias. First, the community averages are calculated over all respondents in the community (not including the index respondent)–not just individuals aged 15 to 24. This recognizes that young people are influenced by all women in their community and not just by those who are in their same age cohort. Second, unlike the peer effects literature, in no case do we use the average of the dependent variable for other individuals as a regressor. Third, over 80% of the respondents have lived in their communities for more than five years and so self-selection which is a source of major concern in the peer effects literature should be a minor issue.

**Simulation procedure.** All of our outcome variables are discrete and we use logit based methods to estimate the parameters in the model. Unfortunately, logit coefficients can only be estimated up to an unknown positive scale factor and so logit coefficients can only be interpreted in terms of sign and significance level since the positive constant does not affect the sign of the coefficient and the scale factor divides out when a t or z statistic is calculated. Therefore, we use simulations to examine the magnitudes of effects–both direct and indirect. In addition, since the only time varying variables in the model are the respondent's age, age squared and her level of education, we use the simulation procedure to "age" each woman one year at a time starting at age 12 until she reaches age 24. This allows us then to predict age at first sex and the pattern of births and method choice as if all 3,454 women aged 15–24 years at the time of the survey were interviewed at age 24 which then allows us to determine how variation in community beliefs and attitudes affect patterns of behavior for each woman over the entire period from age 12 to 24. Finally, because the simulations produce results that are not

scale dependent, we can directly compare results for our corrected model versus a model that is not corrected for selection bias to see how much difference the correction makes. More details on our simulation methodology are presented in the S1 File.

The simulation Fortran program was written by one of the co-authors of this article and was specifically written to do the simulations described in here. The actual simulation program (simulation.f) is included in the S2 File.

## Results

### Descriptive results

Table 1 presents weighted descriptive characteristics and the reproductive behaviors of the sample and shows that 38% of the respondents had first sex at the time of the survey. This is not surprising since our sample consists of 15 to 24 year olds. A total of 19% of respondents had ever given birth and among those who had, the average age at the time of the birth was 18.8 years. Among those who had a birth by the time of the survey, 59% had one birth, 27% had two births, and 14% had more than two births. About 61% of one-year postpartum intervals involved no contraceptive method use. The most frequent contraceptive method choice among those who adopted a method was a traditional method. This was followed by use of a facility-based method (mostly injectable), a shop/pharmacy-based method (mostly condoms) and LAM.

The left side of Table 1 presents descriptive characteristics. For the individual level variables, we see that the average age is 19 and years of education averages almost nine years but this number is censored by the woman's age at the time of the survey. More than 42% of the sample is from Kaduna and the remaining participants are from Ilorin (34%) or Jos (24%). Sixty-two percent of the sample is Muslim while one third (32%) is Protestant and six percent are Catholic; for the analyses below, we compare Muslim to all other religions (Christian). Nearly 80% of the sample reports being strongly or somewhat religious. Finally, 80% of the sample have lived in their current place of residence more than 5 years, 12% have lived in the city 1–5 years and the rest are recent arrivals.

Table 2 presents the weighted community level averages for the attitudes and belief variables among the full sample. These variables are used to construct the indexes for contraceptive norms, provider norms, myth norms, and childbearing norms. Starting with contraceptive norms, we see that more than 53% of the respondents ages 15–49 think that their best friend would approve of FP and almost 45% feel that they would receive praise from members of their community for using contraception. About 13% believe that most or almost all unmarried girls who are sexually active are using contraception and 19% responded that all or most of their friends in the community were using contraception. Finally, nearly 53% of respondents report that they have heard or seen a government official speak in favor of FP in the last year and about 46% had heard or seen the same from a religious leader. There were three variables used in the construction of the provider norm index and they indicate that about 10% of respondents agreed with each of the statements about provider treatment. The final index hypothesized to affect contraceptive method choice is the myth norm index. Eight variables as listed in Table 2 were used to construct the index with all components expected to have an effect in the negative direction on postpartum contraceptive use. We see that 38% of respondents feel that people who use family planning could have health problems and 37% feel that contraceptives can be dangerous to your health. Smaller percentages were associated with various other myths such as family planning causes deformed babies and cancer.

We have one question regarding perceptions about sexual behaviors of unmarried girls and 25% of respondents think that all or most unmarried girls in their community are sexually

active. Finally, three variables are used to construct the childbearing norm index: 36% of respondents say that it is good to have many children, 18% feel the ideal age to get pregnant for the first time is under 20 years of age, and 2.6% feel that ideal child spacing is less than two years.

## Multivariate results

Table 3A and 3B present the results from the joint estimation of our three outcome equations that are corrected for selection and the presence of endogenous regressors. For purposes of comparison, we show uncorrected results in the S1 File. The supporting information also contains the estimation results for the heterogeneity parameters that are strongly jointly significantly different from zero with a p value near zero, indicating the need to use a joint estimation strategy for the three outcome equations.

The results are separated into statistics for individual variables and community beliefs and attitude variables. Table 3A shows the results for the five-category postpartum contraceptive use outcome variable: non-user, traditional method user, using LAM, using a shop/pharmacy-based method, and using a facility-based method. Since we use the multinomial logit model, we estimate the log odds of each of the four methods compared to nonuse for a total of four sets of results. As is well known, any other combination of log odds can be derived from the ones we present. However, since we simulate the predicted probability of use of each method in the next section, we felt that adding additional comparisons was superfluous. All three norms listed in Table 2 (contraceptive, provider, and myth) were initially included in this equation but the provider norm variable was not significant. Since it was correlated with the other two norms, we dropped it from the final model to increase precision.

For the traditional versus nonuse and LAM versus nonuse comparisons, neither norm is significant at any standard level of significance. The results for use of a shop/pharmacy-based method versus no method indicate a positive but not quite significant effect for the contraceptive norm (p = 0.107) but no significant effect for the myth norm. In contrast, the contraceptive norm variable has a positive effect on facility-based method use vs. nonuse (p<0.01) and, in addition, the myth norm has a negative and significant effect (p<0.05). These significant results are in the hypothesized direction. Age at first sex has a negative effect on LAM use versus nonuse (p<0.01) and a positive effect on facility-based method use versus non-use (p<0.10) while the dummies for first and second birth have strong negative effects on the odds of using a facility-based method versus no method in the postpartum period compared to women having higher order births (p-values less than 0.01). Note that the endogeneity of these birth dummies is controlled for by the joint estimation of the method choice equations with the birth hazard equations and that these results for age and first sex and the birth dummies indicate that we should see indirect effects of the norm variables in these equations on contraceptive method choice.

There is one community beliefs and attitudes variable included in the timing of first sex regression: the percentage of respondents in each community that believe that most or all unmarried girls are sexually active (Table 3B). If a greater percentage of members of the community believe that most or all unmarried girls are sexually active, then the probability of initiating sexual activity at each age increases (p<0.05).

In the birth hazard equation, we see that the longer women wait to initiate sexual activity reduces the hazard of having a child at each age after first sex. The fact that this variable is so strongly significant (p<0.001) means that we can expect to see indirect effects of the community beliefs and attitudes variables from the timing of first sex equation on births. The community belief variable is the childbearing norm index constructed by principal components

**Table 3. a. Multivariate results for postpartum contraceptive method choice (Table 3a) and the timing of age at first sex and timing of births (Table 3b) from corrected models, Nigeria, 2017.** b. Multivariate results for postpartum contraceptive method choice (Table 3a) and the timing of age at first sex and timing of births (Table 3b) from corrected models, Nigeria, 2017.

**A.**

| | Method: Traditional vs. Non-use | | | Method: LAM vs. Non-use | | | Method: Shop/pharmacy-based method vs. Non-use | | | Method: Facility-based method vs. Non-use | | |
|---|---|---|---|---|---|---|---|---|---|---|---|---|
| | Corrected Results | | | Corrected Results | | | Corrected Results | | | Corrected Results | | |
| | Coef. | SE | z | Coef. | SE | z | Coef. | SE | z | Coef. | SE | z |
| **Individual Variables** | | | | | | | | | | | | |
| Constant | -8.987 | 2.56 | -3.52*** | 2.518 | 2.35 | 1.07 | -3.248 | 1.68 | -1.93† | -5.099 | 3.50 | -1.46 |
| Age | 0.125 | 0.08 | 1.53 | 0.269 | 0.13 | 1.99* | 0.198 | 0.10 | 1.92† | -0.019 | 0.09 | -0.21 |
| Education | 0.058 | 0.05 | 1.26 | 0.159 | 0.09 | 1.76† | 0.146 | 0.05 | 2.69** | 0.006 | 0.05 | 0.13 |
| Ilorin (ref. Kaduna) | 0.405 | 0.41 | 1.00 | -3.958 | 2.13 | -1.86† | 1.147 | 0.47 | 2.45* | -0.724 | 0.44 | -1.63 |
| Jos (ref. Kaduna) | 0.796 | 0.40 | 2.00* | 0.216 | 0.97 | 0.22 | 0.454 | 0.52 | 0.87 | 0.341 | 0.34 | 0.99 |
| Muslim (ref. non-Muslim) | 0.380 | 0.33 | 1.16 | -0.795 | 0.58 | -1.38 | -0.621 | 0.46 | -1.36 | -0.818 | 0.30 | -2.75** |
| Religiosity (ref. not religious) | 0.014 | 0.32 | 0.05 | -0.162 | 0.57 | -0.29 | -0.045 | 0.37 | -0.12 | 0.042 | 0.26 | -0.16 |
| Lived in city ≤1 yr. (ref. >5 yrs) | -0.039 | 0.44 | -0.09 | 1.012 | 0.79 | 1.29 | -0.021 | 0.64 | -0.03 | -0.313 | 0.48 | -0.65 |
| Lived in city 1–5 yrs (ref. >5 yrs) | 0.277 | 0.40 | 0.69 | 0.930 | 0.80 | 1.17 | 0.776 | 0.41 | 1.87† | 0.296 | 0.31 | 0.95 |
| 1st birth (ref. 3+ order birth) | 0.180 | 0.50 | 0.36 | -1.132 | 0.92 | -1.24 | -0.385 | 0.68 | -0.57 | -1.888 | 0.47 | -4.05*** |
| 2nd birth (ref. 3+ order birth) | 0.364 | 0.39 | 0.94 | -0.810 | 0.71 | -1.40 | -0.266 | 0.57 | -0.47 | -0.878 | 0.33 | -2.65** |
| Age at first sex | 0.105 | 0.14 | 0.77 | -0.620 | 0.18 | -3.52*** | -0.174 | 0.12 | -1.50 | 0.300 | 0.18 | 1.70† |
| **Community Variables** | | | | | | | | | | | | |
| Myths Norm | -0.265 | 0.20 | -1.30 | -0.673 | 0.47 | -1.44 | -0.112 | 0.20 | -0.56 | -0.432 | 0.18 | -2.38* |
| Contraceptive Norm | -0.337 | 0.23 | -1.48 | 0.179 | 0.70 | 0.26 | 0.570 | 0.35 | 1.61 | 0.687 | 0.22 | 3.16** |

**B.**

| | Timing of Age at First Sex | | | Timing of Births | | |
|---|---|---|---|---|---|---|
| | Corrected Results | | | Corrected Results | | |
| | Coef. | SE | z | Coef. | SE | z |
| **Individual Variables** | | | | | | |
| Constant | -27.754 | 1.61 | -17.21*** | -23.021 | 1.71 | -13.44*** |
| Age | 2.329 | 0.17 | 13.82*** | 2.305 | 0.17 | 13.49*** |
| Age Squared | -0.049 | 0.00 | -10.72*** | -0.051 | 0.00 | -11.79*** |
| Education | -0.182 | 0.02 | -7.64*** | -0.047 | 0.01 | -3.84*** |
| Ilorin (ref. Kaduna) | 0.271 | 0.12 | 2.19* | -0.239 | 0.10 | -2.37* |
| Jos (ref. Kaduna | -0.135 | 0.16 | 0.84 | -0.116 | 0.10 | -1.21 |
| Muslim (ref. non-Muslim) | -0.444 | 0.14 | -3.09** | 0.765 | 0.11 | 7.22*** |
| Religiosity (ref. not religious) | 0.256 | 0.10 | 2.57* | -0.200 | 0.09 | -2.89** |
| Lived in city ≤1 yr. (ref. >5 years) | 0.229 | 0.15 | 1.55 | -0.541 | 0.13 | -4.15*** |
| Lived in city 1–5 yrs (ref. >5 years) | 0.521 | 0.13 | 3.91*** | 0.154 | 0.08 | 1.84† |
| Age at first sex | na | na | na | -0.231 | 0.02 | -11.31*** |
| **Community Variables** | | | | | | |
| Childbearing Norm | na | na | na | 0.541 | 0.18 | 3.02** |
| Sexual activity among unmarried girls | 0.629 | 0.26 | 2.38* | na | na | na |

†p < .10

*p < .05

**p < .01

***p<0.001. Each woman may contribute multiple observations based on her sexual experience, birth experience and postpartum contraceptive use patterns.

analysis using the three variables listed in Table 2. We see that, as expected, it has a positive effect on births (p<0.01).

Across the multiple equation models, the effects of the demographic control variables are generally as expected. In the postpartum contraceptive use models, when we included both age

and age squared neither was significant and so we dropped age squared from the equation and found that age is a significant positive predictor for LAM and shop/pharmacy-based method use and higher education increases the log odds of LAM and choosing a shop/pharmacy-based method compared to nonuse. Few other variables are significant predictors other than the city dummies in some comparisons while the Muslim dummy variable has a strong negative effect on facility-based method choice relative to nonuse.

In the timing of first sex equation, the probability of experiencing first sex increases with age but at a slightly decreasing rate and respondents with more years of education have a decreased probability of first sex at each age. The Muslim dummy variable has a very strong negative effect on the odds of initiating sex at any age while more religious individuals have an increased probability of initiating sex at each age. The final individual level variable concerns mobility. We see that respondents who have been residents of their current city for less than a year or between one and five years are more likely to have had first sex at each age relative to longer term residents. This might reflect that the recent movers are women who recently married and moved to live with their partner; these recent movers would be more likely to be sexually experienced. In the timing of birth models, age, education and city of residence behave in the same way that they did in the timing of first sex equation while Muslim flips sign and is associated with increased odds of having a birth at each age. Being religious has a negative effect on births while duration of residence in the current city has a strong negative effect for durations of one year and less and a weakly positive effect for being a resident for one to five years.

## Simulations

The multivariate results show the link between our variables. We find that even after controlling for selection and endogeneity, age at first sex affects both the timing of births and postpartum contraceptive method choice while birth order affects postpartum contraceptive method choice. This means that any significant community belief appearing in the first two equations can have indirect effects on postpartum method use. Rather than present marginal effects for each variable in each equation. We use the simulation procedure described above and in the (S1 File) to quantify the effects of the community attitude and belief explanatory variables. Note that regardless of the respondent's age at the time of the survey, we can simulate the outcomes year by year until she reaches the age of 24. We then examine her simulated age at first sex, her birth history censored at age 24, and postpartum contraceptive method choice after the births of each of her predicted children. The simulation tables then present unweighted averages across the 3,454 respondents. Since the simulation results are unweighted, we present unweighted statistics for the estimation sample for purposes of comparison in the first simulation table. The weighted and unweighted statistics are only slightly different.

The first set of simulations are presented in Table 4 and compare unweighted statistics for the estimation sample at the time of the survey when the respondents were age 19 years on average to simulations based on our multivariate results where we age each respondent from age 12 to age 24. In this table, we show simulations using multivariate results that are uncorrected for endogenous regressors based on Table 1a and 1b in S1 File and results corrected for selection and the presence of endogenous regressors based on Table 3A and 3B. Thus, the uncorrected results only control for selection on observable variables such as age and education while the corrected results also control for unobservable variables that affect the three outcomes.

We see in Table 4 that only 37% of the estimation sample had first sex by the time of the survey with an average age of 17.72 years at the time of first sex. The simulated percentage

**Table 4. Simulation to age 24 for all respondents compared to actual values for the estimation sample.**

| Variable | Estimation Sample (15–24 year olds) | Simulated to age 24 using uncorrected estimates | Simulated to age 24 using corrected estimates |
|---|---|---|---|
| Number reporting first sex | 1,288 | 2,668 | 3,170 |
| Percent reporting first sex | 36.88 | 77.24 | 91.78 |
| | | (0.72) | (0.48) |
| Average age at first sex (years) | 17.72 | 19.12 | 19.24 |
| | | (0.04) | (0.04) |
| Number of births | 1,082 | 2,736 | 3.018 |
| Percent first births | 62.48 | 53.80 | 56.83 |
| | | (0.95) | (0.90) |
| Percent second births | 26.52 | 26.43 | 26.01 |
| | | (0.84) | (0.80) |
| Method choice | | | |
| Percent nonusers | 62.31 | 55.92 | 50.46 |
| | | (0.95) | (0.91) |
| Percent traditional | 11.92 | 13.05 | 9.97 |
| | | (0.64) | (0.55) |
| Percent LAM | 5.04 | 6.54 | 12.36 |
| | | (0.46) | (0.60) |
| Percent shop/pharmacy-based method | 8.96 | 11.11 | 16.14 |
| | | (0.59) | (0.69) |
| Percent facility-based method | 11.85 | 13.38 | 11.07 |
| | | (0.65) | (0.57) |
| N | 3,454 | 3,454 | 3,454 |

Lactational amenorrhea method (LAM); shop/pharmacy-based methods include: male condoms and daily and emergency; facility-based methods include: IUD, implant, and injectables. The values in this table are based on simulated averages in the full sample.

experiencing first sex rises to 77% in the uncorrected results where we age each respondent to 24 and to over 90% when we also control for unobservables in the corrected results column. We see that the uncorrected and corrected results predict a similar age of first sex (about 19) but the corrected model predicts a over 500 additional births by age 24. We see substantial differences in simulated contraceptive method choice between the uncorrected and corrected results where the uncorrected multivariate results substantially overstate nonuse and understate LAM use. The corrected results also predict substantially more shop/pharmacy-based method use but lower levels of facility-based method use.

Fig 1 provides a more visual comparison of the differences in predicted method use based on the uncorrected and corrected parameter estimates and it is clear that the predicted distributions are quite different.

In Table 4, the numbers in parentheses are bootstrap standard errors (see supporting information for more details) that allow us to test to see if there is a statistically significant difference between the method choice distribution for the uncorrected and corrected simulations. A test of the null hypothesis that the two distributions are the same yields a chi squared statistic of 120.8 with 5 degrees of freedom and so the null hypothesis is rejected with a p value of essentially zero. Below, we present additional simulations using the corrected model and do not present standard errors since all the standard errors are very small.

The simulations in Table 5 set the age of first sex to 15 and 19 for each respondent and keep all other variables at their actual values to see the simulated effect of age at first sex on births

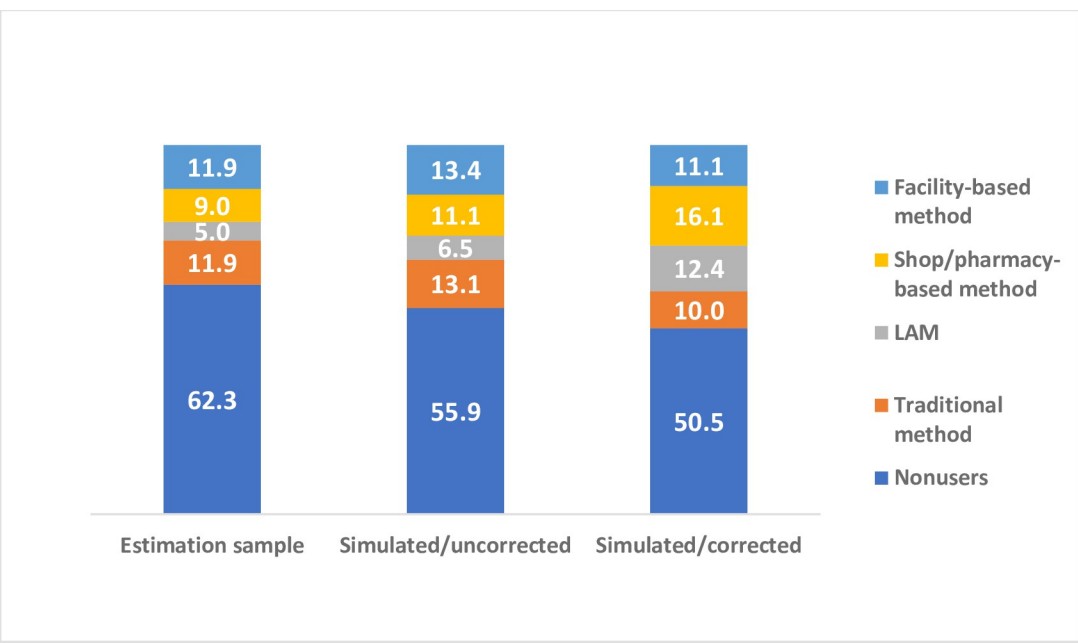

**Fig 1. Postpartum method choice (%) in estimation sample and in simulated samples (uncorrected and corrected).**

and method choice. As one might expect, we see dramatic effects on the simulated number of births. We also see quite different patterns for method choice with individuals initiating sex at age 15 relying much more heavily on LAM in the first year postpartum compared to those initiating sex later. It is interesting that there are similar levels of shop/pharmacy-based method use but those who initiate sex later have almost double the use of facility-based methods.

Table 6 presents simulations for our four community beliefs. For each community belief, we simulate the value from the 10th percentile to the 90th which are extreme cases indicating low levels of beliefs to high levels. For example, the first two columns show what happens to the outcomes if 10% of the community think that all or most unmarried girls are sexually

**Table 5. The simulated effect of age at first sex on births and method choice by age 24 (corrected results).**

|  | Age at first sex 15 | Age at first sex 19 |
|---|---|---|
| **Variable** |  |  |
| Number of births | 7,549 | 3,178 |
| Percent first births | 39.85 | 65.26 |
| Percent second births | 29.30 | 26.49 |
| Method choice |  |  |
| Percent nonusers | 46.48 | 52.99 |
| Percent traditional | 6.46 | 10.23 |
| Percent LAM | 24.76 | 7.86 |
| Percent shop/pharmacy-based method | 15.72 | 15.67 |
| Percent facility-based method | 6.57 | 13.25 |
| N | 3,454 | 3,454 |

Lactational amenorrhea method (LAM); shop/pharmacy-based methods include: male condoms and daily and emergency; facility-based methods include: IUD, implant, and injectables. The values in this table are based on simulated averages in the full sample.

**Table 6. Direct and indirect effects of community beliefs on outcomes (corrected results).**

| Variable | Unmarried girls sexually active (10th percentile) | Unmarried girls sexually active (90th percentile) | Birth Norm (10th percentile) | Birth Norm (90th percentile) | Contraceptive Norm (10th percentile) | Contraceptive Norm (90th percentile) | Myth Norm (10th percentile) | Myth Norm (90th percentile) |
|---|---|---|---|---|---|---|---|---|
| Number reporting first sex | 3,103 | 3,227 | | | | | | |
| % reporting first sex | 89.84 | 92.39 | | | | | | |
| Average age at first sex | 19.37 | 19.11 | | | | | | |
| Number of births | 2,860 | 3,193 | 2,595 | 3,265 | | | | |
| % first births | 56.57 | 55.59 | 60.42 | 55.56 | | | | |
| % second births | 26.15 | 26.15 | 25.32 | 26.31 | | | | |
| Method choice | | | | | | | | |
| % nonusers | 50.35 | 50.77 | 49.71 | 49.16 | 55.63 | 44.37 | 42.54 | 56.99 |
| % traditional | 9.65 | 9.27 | 10.71 | 10.60 | 12.86 | 5.93 | 11.17 | 7.95 |
| % LAM | 13.22 | 13.00 | 12.64 | 12.83 | 12.29 | 12.23 | 17.56 | 9.07 |
| % shop/pharmacy-based | 15.77 | 15.44 | 16.99 | 16.32 | 12.43 | 21.43 | 13.72 | 16.93 |
| % facility-based | 11.01 | 11.52 | 9.94 | 11.09 | 6.79 | 16.03 | 15.01 | 9.04 |
| N | 3,454 | 3,454 | 3,454 | 3,454 | 3,454 | 3,454 | 3,454 | 3,454 |

Lactational amenorrhea method (LAM); shop/pharmacy-based methods include: male condoms and daily and emergency; facility-based methods include: IUD, implant, and injectables. The values in this table are based on simulated averages in the full sample.

active and then if 90% of the community think the same. As this belief moves from the 10th percentile to the 90th, there is a direct effect that decreases the age at first sex in the simulated population (from 19.37 years to 19.11 years) which leads to an increase of 333 in the number of births, a substantial indirect effect. However, postpartum contraceptive method use is similar between the two groups indicating that indirect effects for this community belief on method choice are not substantial.

Table 6 also presents simulations for the childbearing norm that directly affects the number and timing of births and has an indirect effect on postpartum contraceptive method choice. This table also includes the direct effects of the myths and contraception norms on postpartum contraceptive method choice. In the simulation when we compare 10% of community members agreeing with the childbearing norm versus 90% agreeing with the childbearing norm, we find the expected results such that where there is higher agreement there is 670 additional births but there are little in the way of indirect effects on postpartum contraceptive method choice. On the other hand, we see very substantial direct effects on postpartum contraceptive method choice for both contraceptive and myths norms. Where only 10% of the community agrees with the contraceptive norm, nonuse is over 10 percentage points higher and shop/pharmacy-based method use and facility-based method use are lower compared to when 90% of the community are simulated to agree with the contraceptive norm. We expect the myth

norm to have a negative effect on postpartum contraceptive use and that is exactly what we find. As we move from the 10th to the 90th percentile, nonuse increases by about 14 percentage points with the bulk of that decrease coming at the expense of the use of facility-based methods and LAM. We do get one anomalous result where the use of shop/pharmacy-based methods increases slightly.

## Discussion

We found large and significant direct effects of our four community variables on our three outcomes and all effects were in the expected direction. In particular, contraceptive norms and myths are important factors that influence young women's postpartum contraceptive use and choice of a method. Where there are widespread concerns about contraceptive safety and perspectives on who should or should not use contraception (components of the myth norm), young women are less likely to use a contraceptive method postpartum and when they do choose a method, they are less likely to choose a facility-based method (i.e., injectable) that may be more effective for spacing their next birth. Further, where the feeling is that more young people are using contraception and it is more accepted by friends, family, and key decision-makers, there is higher postpartum contraceptive method use, including use of shop/pharmacy-based and facility-based methods. We also found that when age at first sex is later, there are substantially fewer births and an increase in the use of facility-based methods postpartum. However, for the most part, the magnitude of the indirect effects was small relative to direct effects.

This study contributes to earlier studies on postpartum contraceptive use. Prior studies often overlook young mothers as part of family planning programming because of societal pressures for them to marry and demonstrate their fecundability [7]. Further, young women are often not included in the analysis samples for postpartum contraceptive use because of small sample sizes or the common perspective that they do not have contraceptive needs. In addition, previous reviews have indicated the importance of examining cultural norms, gender norms, attitudes, and beliefs that affect young people's ability to act on their postpartum contraceptive intentions and decisions [9, 17, 22]. In many cases, these multiple levels of influence are difficult to study because of problems with selection bias and unobserved measures. This study that included a large sample of young women and used information on community beliefs and attitudes is able to overcome some of these challenges by examining the direct and indirect effects of community beliefs and attitudes on adolescent and youth ages 15–24 postpartum contraceptive method choice. These findings around community beliefs and attitudes and their influence on sexual and reproductive health behaviors, including sexual initiation, timing of early births, and postpartum contraceptive use are useful for considering strategies to strengthen FP programming. While the community beliefs and attitudes measured here do not directly capture community norms, they are representative of community perspectives. Programs seeking to increase contraceptive method use need to identify approaches that can affect these perceptions. This could happen through the use of mass media (e.g., television or radio) whereby young couples are shown deciding to delay a first birth or community leaders are speaking positively about contraception. Likewise, interpersonal communication activities could be launched which engage community or religious leaders to speak to their communities or congregations about the benefits of contraception. Identifying approaches that reach communities with positive messages from trusted sources will be important for influencing community attitudes and beliefs around FP.

This paper is not without limitations. First, the data used in the analysis of postpartum contraceptive use as well as to identify births in the study period come from a retrospective

calendar for the five years before the survey. Calendar data are useful because they obtain a wealth of information; however, they have a risk of respondent recall errors, of heaping on specific dates, and of misreporting of key events [46–48]. Another potential bias in our analysis is that not all the sample had attained the events of interest–first sex and a birth–by the time of the survey.

To conclude we find that addressing community attitudes and beliefs around sexual and reproductive health behaviors of young people, including the timing of first sex, birth intervals, and the acceptability of contraceptive use can lead to improvements in contraceptive behaviors among young people, including increasing postpartum contraceptive use. Strategies to increase postpartum contraceptive use among young people can help to reduce rapid, repeat pregnancies and lead to improved health and well-being for young women, their children, and their communities.

## Supporting information

**S1 File. Supporting methodological information and additional tables.**
(DOCX)

**S2 File. Simulation and Fortran programs to replicate analyses.**
(DOCX)

## Acknowledgments

The authors acknowledge those who supported this work at various phases of project design, implementation, and analysis. This includes: the Nigerian Urban Reproductive Health Initiative (NURHI) Team, data collectors, and study participants.

## Author Contributions

**Conceptualization:** David K. Guilkey, Ilene S. Speizer.

**Formal analysis:** David K. Guilkey, Ilene S. Speizer.

**Methodology:** David K. Guilkey.

**Writing – original draft:** David K. Guilkey, Ilene S. Speizer.

**Writing – review & editing:** David K. Guilkey, Ilene S. Speizer.

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
