## [Decision Letter · Decision Letter 0]

25 Jun 2021

PONE-D-21-18360

The Direct and Indirect Effects of Community Beliefs and Attitudes on Postpartum Contraceptive Method Choice among Young Women Ages 15-24 in Nigeria

PLOS ONE

Dear Dr. Speizer,

Thank you for submitting your manuscript to PLOS ONE. After careful consideration, we feel that it has merit but does not fully meet PLOS ONE’s publication criteria as it currently stands. Therefore, we invite you to submit a revised version of the manuscript that addresses the points raised during the review process.

Two reviewers have provided their feedback providing specific points that can be improved or should be clarified.

In addition, the academic editor feels that the article requires a clarification of the purpose of the article and a justification for the methods adopted in terms of that purpose. This is a complex article, potentially of impact. It is important to highlight what is its specific contribution.

The stated purpose is “Understanding what factors influence postpartum contraceptive use among young people”. Regarding methods the stated purpose is “our statistical methods control for the endogenous timing of the initiation of sexual activity and the timing and number of births to each respondent by simultaneous estimation of equations for these choices with the choice of postpartum contraceptive method”.

However, note that postpartum contraceptive use, as the name states, is only decided postpartum. And postpartum, initiation of sexual activity and the timing of births are given. These do not seem simultaneous decisions requiring correction for endogeneity. They are sequential decisions and the relevant public health dimension seems whether and who adopts the methods postpartum, conditional that they bore children. On the other hand, you are also allowing for correlation in the error terms of the three equations to take into account unobserved heterogeneity and discuss how you have dealt with the social reflection problem. You should provide a clear rationale of why the simulation exercise and this specific application. Are we really interested in the “state” at age 24? Or in a population like the current population?

The justification could be the desire of specific interventions regarding community beliefs and attitudes but this should be made clearer and more specific. Also, any links to literature facing a similar problem with similar methods

Finally, only the joint estimates are discussed. The uncorrected estimates are provided in the supplementary file. This is correct according to PLOS ONE statistical reporting that requires any estimates used to be included as supplements. It would be helpful, however, if you could present graphically and discuss differences in the coefficients of the two models noting that most research on these aspects would stop at the estimation of this equation.

I understand this is related to what you are doing in table 4. However, table 4 looks at a particular distribution at age 24 as long as I understand. BTW: You mention in the heading of table 4 “Simulation to age 24 for all respondents compared to actual values for the estimation sample and to individuals age 24 or 25 in full sample” but the columns only state “Estimation Sample (15-24 year olds)”. Is not a column on “Estimation sample (24-25 year olds” missing?

You are also not providing a rationale for the specific choice of control variables in the three equations, including the use of interactions. Please do so indicating its connection with the relevant literature.

We look forward to receiving your revised manuscript.

Kind regards,

José Antonio Ortega, Ph.D.

Academic Editor

PLOS ONE

Journal Requirements:

2.  Please include additional information regarding the survey or questionnaire used in the study and ensure that you have provided sufficient details that others could replicate the analyses. For instance, if you developed a questionnaire as part of this study and it is not under a copyright more restrictive than CC-BY, please include a copy, in both the original language and English, as Supporting Information. If the original language is written in non-Latin characters, for example Amharic, Chinese, or Korean, please use a file format that ensures these characters are visible."

3. Please state whether you validated the questionnaire prior to testing on study participants. Please provide details regarding the validation group within the methods section."

4. Please amend your current ethics statement to address the following concerns:

a) Did participants provide their written or verbal informed consent to participate in this study?

Reviewers' comments:

Reviewer's Responses to Questions

**Comments to the Author**

1. Is the manuscript technically sound, and do the data support the conclusions?

Reviewer #1: Yes

Reviewer #2: Yes

2. Has the statistical analysis been performed appropriately and rigorously? 

Reviewer #1: Yes

Reviewer #2: Yes

3. Have the authors made all data underlying the findings in their manuscript fully available?

Reviewer #1: Yes

Reviewer #2: Yes

4. Is the manuscript presented in an intelligible fashion and written in standard English?

Reviewer #1: Yes

Reviewer #2: No

5. Review Comments to the Author

Reviewer #1: They need to provide some more seccondary data of their main research proplems at introduction section to make an argument stronger than the present one. In addition, they have to describing main purpose of simulation process i.e. for what reason. And it is very important to discuss the main result after focus on influences of community factors, and simulation outcome.

Reviewer #2: This is an interesting paper; however, might require major revision and editing before it could be published. Please consider addressing the given below points:

Abstract

- Line 41: Maybe it would be ‘We found’ instead of ‘We find’?

- Line 42-43: Are all 3 specified outcomes (sexual initiation, birth timing, and postpartum contraceptive use) the primary outcomes? Based on the title of the study, postpartum contraceptive use seems to be the primary outcome of the study.

Introduction

- Page 4, Line 79: Need a citation

- Page 4, Line 85: Consider including some examples related to norms, attitudes, and beliefs that the authors believe would influence young people’s reproductive health behaviors.

- Young mothers have been often included in the studies of postpartum contraceptive use. This is in contrast to what authors have specified.

Methods

- Line 90-97: Details of the study context could be specified under the ‘Introduction’ section instead of ‘Methods’ section. This would allow readers to understand the reason why Nigeria was selected for the conduct of this study ahead of the methods applied.

- Line 113: What was the duration for Phase 2 (i.e., 2015- xx)?

- Table 1: Consider including % for other religions as well. Specify what were included in the traditional methods. Generally, IUDs and implants are considered as long-acting methods. Did the survey include female sterilization too under the long-acting methods rather than the category of permanent methods?

- Not sure why the authors decide to include Table 1 under the methods section as opposed to the standard results section.

- Line 145-147: Might require paraphrasing. This sentence is a bit confusing to me.

- Line 155-156: The definitions used for the short acting and long-acting methods is not the same as specified under the footnote of Table 1.

- In Table 1, there are too many dependent/outcome variables listed, which is in contrast to the title of the study as well as what has been included as a background information under the ‘Introduction’ section.

- Line 180: Write the full form for PSU as it’s used for the first time in the text.

- Table 2: Provider norms was collinear with which other contraceptive belief variables? This needs to be specified.

- Line 208: Why age 12 to 24 and not 15 to 24?

- I am not quite sure if the inclusion of details of all the equations were required for the purpose of this study.

- Why 1-year postpartum was used in this study when the authors have discussed about the requirement of 3-year birth-interval? This needs to be clear under the methods section.

- Overall, please consider writing the methods section in the past tense.

Results

- Methods and results are so mixed up that its hard to follow the text. First clearly specify what statistical analyses were used and for what reason. This could be followed by the results.

- Consider including only the key study findings rather than describing all the findings that’s also presented in the Tables.

- Many terms for the types of contraception have been used throughout the paper. For e.g., modern contraception, effective contraception etc. I would recommend authors to be very specific in using these terms Also, all these terms need to be defined clearly with a proper citation. The reason for doing so would be the use of different definitions based on the available literatures on family planning.

Discussion

- First paragraph of the discussion: There is a repetition of methods here again under the discussion section. Consider avoiding it.

- Discussion is not adequate and needs elaboration. Would recommend the authors to follow the following pattern to draft a discussion section:

o Key findings in relation to the research question stated.

o Interpretations of the findings.

o Comparison and contrast of study findings.

o Strengths and limitations of the study.

o Implications of the study findings.

6. PLOS authors have the option to publish the peer review history of their article (what does this mean?). If published, this will include your full peer review and any attached files.

Reviewer #1: **Yes: **Yothin Sawangdee

Reviewer #2: No

---

## [Author Response · Author response to Decision Letter 0]

27 Jul 2021

Response: Note about the tables – Because we made some minor modifications to the models, the multivariate tables and simulation results all changed. For the tables presented, we only show the final models and do not track changes since that is complicated for reviewing the materials. In Table 1, we have added additional details on religion and on the methods that make up the postpartum method use variable. This information again is not tracked to simplify the table but we did highlight the variables that are added with more detail. Also, in Table 1, we relabeled the column on the right to be Reproductive Variables and we bolded our primary outcome of postpartum method choice.

Response: For the reference list, we have highlighted references that were newly added to the list.

Academic Editor Comments

Overall comment: The academic editor feels that the article requires a clarification of the purpose of the article and a justification for the methods adopted in terms of that purpose. This is a complex article, potentially of impact. It is important to highlight what is its specific contribution.

Response: We have revised the discussion and methods sections based on detailed inputs from the academic editor and reviewers to help provider greater context for the contribution of the paper and to justify analysis methods. 

The stated purpose is “Understanding what factors influence postpartum contraceptive use among young people”. Regarding methods the stated purpose is “our statistical methods control for the endogenous timing of the initiation of sexual activity and the timing and number of births to each respondent by simultaneous estimation of equations for these choices with the choice of postpartum contraceptive method”.

However, note that postpartum contraceptive use, as the name states, is only decided postpartum. And postpartum, initiation of sexual activity and the timing of births are given. These do not seem simultaneous decisions requiring correction for endogeneity. They are sequential decisions and the relevant public health dimension seems whether and who adopts the methods postpartum, conditional that they bore children. 

Response: We agree with the reviewer that these are sequential decisions and not simultaneous. However, if there are unobservable variables that affect the three decisions, then selection bias is introduced if one does not account for it. For example. The respondent may have knowledge of her fecundity which could affect the timing of first sex, the timing of births and postpartum method choice. The seminal work on this type of selection bias caused by common unobservables was due to Heckman (1979) -- work for which he won the Nobel Prize in Economics. His application was the determination of wage rates for employed individuals. Clearly before one can observe a wage rate, the individual must first be employed and if common unobservables affect both the decision to work and then the wage rate conditional on working, the coefficients of the wage equation are biased. For example, more highly motivated individuals may be more likely to accept a job and may be observed with a higher wage rate conditional on being employed.

Selection bias in hazard models due to unobservables, such as our equations for the timing of first sex and timing of births, are especially problematic as demonstrated by Heckman and Singer (1984) in a set of Monte Carlo experiments. In addition, the methods we use in this paper have frequently been used in fertility related research. See, for example, Angeles, Guilkey and Mroz (2005) for a study of the fertility and female education in Indonesia; Rindfuss, Guilkey, Morgan, and Kravdahl (2007) for a study of the effects of the availability of childcare on fertility in Norway; and Lillard (1993) for a study of marriage duration and fertility timing in the U.S. Further, Lillard and Panis (2003) also wrote a statistical software package called aML that can be used to simultaneously estimate hazards models. Our software is similar to theirs except that they assume multivariate normality for the error terms while we use a non-parametric approach.

We now provide a fuller justification for our statistical methods based on this response.

Angeles, G. D. Guilkey, and T. Mroz 2005. “The Effects of Education and Family Planning Programs on Fertility in Indonesia: Economic Development and Cultural Change Volume 54:1, pp.165-201.

Heckman, J. 1979. “Sample Selection Bias as a Specification Error” Econometrica Lomune 47:1, pp. 153-161.

Heckman, J. and B. Singer 1984. “ A Method for Minimizing the Impact of Distributional Assumptions in Econometric Models for Duration Data” Econometrica Volume 52:2, pp. 271-320.

Lillard, L. and C. Panis 2003. “aML Multilvel Process Statistical Software Version 2.0. Los Angeles: Econware.

Lillard, L. 1993. “Simulataneous Equations for Hazards: Marriage Duration and fertility Timing” Journal of Econometrics Volume 56:1-2, pp. 189-217.

Rindfuss, R. D. Guilkey, P. Morgan, O. Karvdal. And K. Guzzo 2007. “Child care availability and first-birth timing in Norway”, Demography 44:2,pp.345-372.

On the other hand, you are also allowing for correlation in the error terms of the three equations to take into account unobserved heterogeneity and discuss how you have dealt with the social reflection problem. You should provide a clear rationale of why the simulation exercise and this specific application. Are we really interested in the “state” at age 24? Or in a population like the current population?

Response: It is the correlation in the error terms that causes the selection bias and we provide an explicit test of this correlation. We reject the null hypothesis of no correlation at all standard levels of significance which is the main justification for our estimation strategy. The result of the test means that uncorrected results are biased and we can see the size of the bias by comparing simulations that use uncorrected and corrected estimated parameters.

Another rationale for the simulation exercise is that all of our outcome variables are discrete and we use logit based methods to estimate the parameters in the model. Unfortunately, logit coefficients are only known up to an unknown positive scale factor and so logit coefficients can only be interpreted in terms of sign and significance level since the positive constant does not affect the sign of the coefficient and the scale factor divides out when a t or z statistic is calculated. The simulations, on the other hand, allow us to examine the magnitudes of effects. For example, we can see the effect of a specific age at first sex on the probability the individual uses each method type in the year after the birth of each child.

Our simulation is not actually meant to say what a cohort of 24 year olds look like but rather, how the predicted experience of a cohort will look when they pass through their young adult years using the model’s parameters to form the predictions. In other words we see predicted behavior at every age from 12 to 24.

We have clarified the rationale for the simulation exercise in the paper and we have tried to strengthen our reasoning for focusing on respondents under the age of 25. 

The justification could be the desire of specific interventions regarding community beliefs and attitudes but this should be made clearer and more specific. Also, any links to literature facing a similar problem with similar methods

Response: We have clarified above that the approach we are using is helping to determine which of the community beliefs and attitudes are related to postpartum method use either directly or indirectly. We have extended our background and methods sections to make this clearer in the current version of the paper. We have also added additional links to the literature – see the references listed above in our response to justify our statistical approach.

Finally, only the joint estimates are discussed. The uncorrected estimates are provided in the supplementary file. This is correct according to PLOS ONE statistical reporting that requires any estimates used to be included as supplements. It would be helpful, however, if you could present graphically and discuss differences in the coefficients of the two models noting that most research on these aspects would stop at the estimation of this equation.

Response: As discussed above, one cannot directly compare estimated coefficients from the estimation of logit based models and so we could only compare the corrected and uncorrected coefficients in terms of sign and significance. We could certainly do that if the editor would like us to. However, in Table 4, we present corrected and uncorrected simulation results based on our multivariate results that can be directly compared. We see that the uncorrected results overstate the percentage of nonusers, understate the percentage using shop/pharmacy-based methods and overstate the percentage using facility-based methods.

In addition, we tested to see if the predicted distributions of method use were statistically different for the corrected versus uncorrected simulations. The null hypothesis that the distribution was the same was rejected at all standard levels of significance with a p value of essentially zero and one must conclude that models that do not correct for selection bias give misleading results. We provide details on how this test was done in the text.

In response to the reviewer’s suggestion, we have added a graph (see Figure 1) which compares the simulated distribution of method use from the correct and uncorrected results and strengthened our discussion for the need to use the corrected results.

I understand this is related to what you are doing in table 4. However, table 4 looks at a particular distribution at age 24 as long as I understand. 

Response: The distributions of method use are averages of postpartum method use after each simulated birth that a respondent has from her first birth at whatever age that occurred up until her last birth by age 24. We have tried to make it clear that we are looking at postpartum method use after the birth of each child to the mother between the ages of 12 and 24.

BTW: You mention in the heading of table 4 “Simulation to age 24 for all respondents compared to actual values for the estimation sample and to individuals age 24 or 25 in full sample” but the columns only state “Estimation Sample (15-24 year olds)”. Is not a column on “Estimation sample (24-25 year olds” missing?

Response: We thank the reviewer for pointing this out. The table was mislabeled and we have corrected the title.

You are also not providing a rationale for the specific choice of control variables in the three equations, including the use of interactions. Please do so indicating its connection with the relevant literature.

Response: We have revised the text and now include references that relate to the list of control variables. In this revised version of the paper, we have removed the interactions that we had previously included to simplify the presentation and discussion since they were not central to the questions we are trying to answer; the removal of these interactions did not significantly change the results. 

Journal Requirements:

Response: We have made the relevant changes to the PLOS ONE format.

2. Please include additional information regarding the survey or questionnaire used in the study and ensure that you have provided sufficient details that others could replicate the analyses. For instance, if you developed a questionnaire as part of this study and it is not under a copyright more restrictive than CC-BY, please include a copy, in both the original language and English, as Supporting Information. If the original language is written in non-Latin characters, for example Amharic, Chinese, or Korean, please use a file format that ensures these characters are visible."

Response: The questionnaires and data are all available on our dataverse site and this is mentioned in the data availability statement and in the main text. 

3. Please state whether you validated the questionnaire prior to testing on study participants. Please provide details regarding the validation group within the methods section."

Response: We provide information in the text that we used validated questionnaires from the Measurement, Learning & Evaluation project. 

4. Please amend your current ethics statement to address the following concerns:

a) Did participants provide their written or verbal informed consent to participate in this study?

Response: Verbal consent was used and the consent form was signed by the interviewer. This has been added to the text.

Response: This is the consent procedure that has been used by large global surveys (e.g., the Demographic and Health Survey) as well as by the earlier rounds of data collection by the Measurement, Learning & Evaluation (MLE) project. This was a follow-on data collection effort in two of the sites from the MLE project. All consent procedures were approved by the in-country and UNC IRB committees as indicated in the text. 

Response: These modifications have been made to the current version of the paper.

Reviewers' comments:

Reviewer's Responses to Questions

Comments to the Author

1. Is the manuscript technically sound, and do the data support the conclusions?

Reviewer #1: Yes

Reviewer #2: Yes

2. Has the statistical analysis been performed appropriately and rigorously? 

Reviewer #1: Yes

Reviewer #2: Yes

3. Have the authors made all data underlying the findings in their manuscript fully available?

Reviewer #1: Yes

Reviewer #2: Yes

4. Is the manuscript presented in an intelligible fashion and written in standard English?

Reviewer #1: Yes

Reviewer #2: No

Response: We have reviewed the text and attempted to address any typographical and grammatical errors. 

5. Review Comments to the Author

Reviewer #1: They need to provide some more secondary data of their main research problems at introduction section to make an argument stronger than the present one. In addition, they have to describing main purpose of simulation process i.e. for what reason. And it is very important to discuss the main result after focus on influences of community factors, and simulation outcome.

Response: We have revised the relevant sections of the paper to address these issues. This includes revisions to the introduction, the discussion of the simulation methods, as well as a more complete discussion section. 

Reviewer #2: This is an interesting paper; however, might require major revision and editing before it could be published. Please consider addressing the given below points:

Abstract

- Line 41: Maybe it would be ‘We found’ instead of ‘We find’?

Response: This abstract is written in the present tense and thus, we prefer to leave it as “we find.” 

- Line 42-43: Are all 3 specified outcomes (sexual initiation, birth timing, and postpartum contraceptive use) the primary outcomes? Based on the title of the study, postpartum contraceptive use seems to be the primary outcome of the study.

Response: We have revised the abstract and text to clarify that postpartum contraceptive use is the primary outcome. 

Introduction

- Page 4, Line 79: Need a citation

Response: Citations and additional framing of the research problems have been added to this section based on this and the previous reviewer’s comments. 

- Page 4, Line 85: Consider including some examples related to norms, attitudes, and beliefs that the authors believe would influence young people’s reproductive health behaviors.

Response: Examples have been added as requested.

- Young mothers have been often included in the studies of postpartum contraceptive use. This is in contrast to what authors have specified.

Response: We have revised the introduction to add in more studies that address young mothers’ contraceptive needs. 

Methods

- Line 90-97: Details of the study context could be specified under the ‘Introduction’ section instead of ‘Methods’ section. This would allow readers to understand the reason why Nigeria was selected for the conduct of this study ahead of the methods applied.

Response: This has been moved to the end of the introduction section as a separate section. 

- Line 113: What was the duration for Phase 2 (i.e., 2015- xx)?

Response: The NURHI Phase 2 project was 2015-2020 which has been added to the text.

- Table 1: Consider including % for other religions as well. Specify what were included in the traditional methods. Generally, IUDs and implants are considered as long-acting methods. Did the survey include female sterilization too under the long-acting methods rather than the category of permanent methods?

Response: We have added the requested clarifications to Table 1 to address these issues. 

- Not sure why the authors decide to include Table 1 under the methods section as opposed to the standard results section.

Response: We have included it here as it helps with describing the sample and the variables. Thus, it felt more appropriate in the methods section for this paper.

- Line 145-147: Might require paraphrasing. This sentence is a bit confusing to me.

Response: We have revised this statement and hope that the revised version is clearer.

- Line 155-156: The definitions used for the short acting and long-acting methods is not the same as specified under the footnote of Table 1.

Response: Good catch, we have fixed this and revised the labeling and definition to be more appropriate. We now talk about shop/pharmacy-based methods (pills, EC, condoms) and facility-based methods (injections, IUD, implant). 

- In Table 1, there are too many dependent/outcome variables listed, which is in contrast to the title of the study as well as what has been included as a background information under the ‘Introduction’ section.

Response: Thank you for this comment. We have revised the table to clarify that the variables on the right are reproductive health related variables; we have bolded the primary outcome of postpartum method use. 

- Line 180: Write the full form for PSU as it’s used for the first time in the text.

Response: Good catch, this change has been made.

- Table 2: Provider norms was collinear with which other contraceptive belief variables? This needs to be specified.

Response: This has now been specified in the footnote of the table.

- Line 208: Why age 12 to 24 and not 15 to 24?

Response: The reason that we start at age 12 instead of age 15 is that some respondents reported age at first sex prior to age 15 and by going back to an earlier age, we eliminate left censoring from the hazard model. This has now been clarified in the text where we describe the analysis methods.

- I am not quite sure if the inclusion of details of all the equations were required for the purpose of this study.

Response: We have included the equations to help a reader to follow the complex analyses and kept them in since the editor did not seem to think this needed to be removed. They could be moved to the supplementary information if needed. 

- Why 1-year postpartum was used in this study when the authors have discussed about the requirement of 3-year birth-interval? This needs to be clear under the methods section.

Response: While three-year birth intervals are the target, the postpartum period is usually considered the first year after a birth and this is a crucial time to get a woman to adopt a contraceptive method. We have explained this in the text. 

- Overall, please consider writing the methods section in the past tense.

Response: We have tried to be consistent throughout the paper in the tense used. 

Results

- Methods and results are so mixed up that its hard to follow the text. First clearly specify what statistical analyses were used and for what reason. This could be followed by the results.

Response: We have tried to simplify the presentation of the methods and results and hopefully it is clearer now. 

- Consider including only the key study findings rather than describing all the findings that’s also presented in the Tables.

Response: We have simplified the presentation of the findings as suggested by this reviewer. 

- Many terms for the types of contraception have been used throughout the paper. For e.g., modern contraception, effective contraception etc. I would recommend authors to be very specific in using these terms Also, all these terms need to be defined clearly with a proper citation. The reason for doing so would be the use of different definitions based on the available literatures on family planning.

Response: We have gone through the paper and made an effort to be more consistent with the terminology used.

Discussion

- First paragraph of the discussion: There is a repetition of methods here again under the discussion section. Consider avoiding it.

- Discussion is not adequate and needs elaboration. Would recommend the authors to follow the following pattern to draft a discussion section:

o Key findings in relation to the research question stated.

o Interpretations of the findings.

o Comparison and contrast of study findings.

o Strengths and limitations of the study.

o Implications of the study findings.

Response: Thank you, we have revised the discussion section based on these inputs and inputs from the previous reviewer.

---

## [Editor Report · Decision Letter 1]

18 Nov 2021

PONE-D-21-18360R1The Direct and Indirect Effects of Community Beliefs and Attitudes on Postpartum Contraceptive Method Choice among Young Women Ages 15-24 in NigeriaPLOS ONE

Dear Dr. Speizer,

Thank you for submitting your manuscript to PLOS ONE. After careful consideration, we feel that it has merit but does not fully meet PLOS ONE’s publication criteria as it currently stands. Therefore, we invite you to submit a revised version of the manuscript that addresses the points raised during the review process. The manuscript has considerably strengthened and is much clearer for a broad general audience such as PLOS ONE's. Congratulations to the authors. It is not felt necessary to send the article back to the reviewers. 0. There are, however, some details that are left unspecified and should be addressed according to PLOS ONE statistical guidelines. You mention in the reply that you are using non-standard software but no mention is given in the text, or any links to code for reproducibility. These are the specific PLOS ONE guidelines on the topic: 

"In the methods, include a section on statistical analysis that reports a detailed description of the statistical methods. In this section:

List the name and version of any software package used, alongside any relevant referencesDescribe technical details or procedures required to reproduce the analysisProvide the repository identifier for any code used in the analysis (See our code-sharing policy.)"

Please update.

Regarding interpretation of results,  and details regarding the description of results I would ask for some clarifications:

1. Please clarify on the time unit of analysis. Is it monthly and annual? It seems that the calendar data is monthly but at some time it is said that age is t+13 suggesting estimation is based on annual data.

2. How is method use defined in the presence of concurrent method use? Did you contemplate an alternative state for concurrent method use? Do you use the most effective method à la DHS?

3. Related to that, if estimation is carried out with yearly and not monthly data: How do you define the method used in a year time? Any use? Most frequent method?

4. This is particularly relevant in the case of LAM.

Maybe you could clarify that the woman might be using LAM without any intention to contracept. To the extent that the conditions of not menstruating, baby less than 6 months, and full breast-feeding are met the woman is using LAM. In this sense it is very different from the other methods (but you are allowing for this. You could comment in the manuscript).Also, you cannot be using LAM for more than 6 months. It is therefore not a solution as a post-partum method (just for the immediate postpartum). I would like to know how this is captured in the model, particularly in connection with questions 2 and 3.

We look forward to receiving your revised manuscript.

Kind regards,

José Antonio Ortega, Ph.D.

Academic Editor

PLOS ONE
---

## [Author Response · Author response to Decision Letter 1]

3 Dec 2021

Editor comments :

Thank you for submitting your manuscript to PLOS ONE. After careful consideration, we feel that it has merit but does not fully meet PLOS ONE’s publication criteria as it currently stands. Therefore, we invite you to submit a revised version of the manuscript that addresses the points raised during the review process.

The manuscript has considerably strengthened and is much clearer for a broad general audience such as PLOS ONE's. Congratulations to the authors. It is not felt necessary to send the article back to the reviewers.

There are, however, some details that are left unspecified and should be addressed according to PLOS ONE statistical guidelines. You mention in the reply that you are using non-standard software but no mention is given in the text, or any links to code for reproducibility. These are the specific PLOS ONE guidelines on the topic:

"In the methods, include a section on statistical analysis that reports a detailed description of the statistical methods. In this section:

• List the name and version of any software package used, alongside any relevant references

• Describe technical details or procedures required to reproduce the analysis 

• Provide the repository identifier for any code used in the analysis (See our code-sharing policy.)" 

Please update. 

RESPONSE:

We have added relevant information into the statistical methods section on the estimation approaches and software used. In addition, we have added an additional supporting information file (S2) that includes the code for reproducing the analysis in Fortran. As we elaborate in the supporting information file, a total of three programs were used:

1. The Discrete Factor Approximation Method (DFAM)

This program estimates systems of equations with a mix of continuous and discrete dependent variables using semi-parametric maximum likelihood. It was written and copyrighted by Jeffrey Rous from the Department of Economics at North Texas State University. The manual for the package (referred to as LEO) describes the program, specifies the likelihood function and provides detailed instructions on how to write a setup file that the program uses to specify the set of equations and read in the raw data as an ascii file. A copy of the manual has been put included as part of the supporting information file (S2). The actual DFAM program code would have to be obtained from Jeffrey Rous and we have added this note to the text.

2. GQOPT

This is the Goldfeld-Quandt non-linear optimization package (www.quandt.com/gqopt.html) that is marketed by Richard Quandt from Princeton University. The package includes a large number of non-linear optimization packages. The DFAM (LEO) program can invoke two of the packages: the Davidon-Fletcher-Powell (DFP) algorithm and the quadratic hill climbing algorithm (GRADX). We used DFP as a first step and then switched to GRADX when we got close to the maximum of the likelihood function to get more precision and a better estimate of the covariance matrix of the parameter estimates. In both cases, we used the default versions of the algorithms; this is described in the text and supporting information file (S2).

 3. Simulation Program

The simulation program was written in Fortran 77 by one of the co-authors of this article and was specifically written to do the simulations described in detail in the text and the supporting information files (S1 and S2). 

DFAM (LEO) was provided to us by Rous under the provision that we not share his copyrighted code. We refer readers to Rous at North Texas State University to obtain a copy. A site license for GQOPT was purchased by the Department of Economics at the University of North Carolina at Chapel Hill for use by our faculty and graduate students. We refer readers to Quandt to obtain a copy of the code. The simulation program (simulation.f) and the setup file for DFAM (s.fil) which were written by one of the co-authors of this article are included in supporting information file S2. 

Additional Comments: 

Regarding interpretation of results, and details regarding the description of results I would ask for some clarifications:

1. Please clarify on the time unit of analysis. Is it monthly and annual? It seems that the calendar data is monthly but at some time it is said that age is t+13 suggesting estimation is based on annual data. 

RESPONSE: This analysis was performed annually such that we examined use in the one year following a birth and examined the most effective method used in that year. Therefore, each woman contributes as many observations as births that she has in the calendar period. This has been clarified in the text.

2. How is method use defined in the presence of concurrent method use? Did you contemplate an alternative state for concurrent method use? Do you use the most effective method à la DHS? 

RESPONSE: In the calendar, only one method can be reported per month; interviewers were instructed to report the most effective method in that month if multiple methods were reported. This has been added to the text.

3. Related to that, if estimation is carried out with yearly and not monthly data: How do you define the method used in a year time? Any use? Most frequent method? 

RESPONSE: Most people only listed one method in the year postpartum. For the small number of women who used multiple methods in the year, we used the most effective method listed in the period. We consider long-acting methods the most effective followed by short acting methods. This has been clarified in the text.

4. This is particularly relevant in the case of LAM. 

RESPONSE: The order for effectiveness was none, traditional, LAM, short acting, long acting. If they simply reported LAM, then that was the method that they were assigned in the postpartum period. However, if they switched to a long-acting or short acting method during the year, then they were assigned that method rather than LAM. There are only 54 individuals that reported uniquely LAM use. Note that this use is coded based on use of a method (the most effective method) at some point during the year postpartum; all of the LAM use started immediately postpartum. 

Maybe you could clarify that the woman might be using LAM without any intention to contracept. To the extent that the conditions of not menstruating, baby less than 6 months, and full breast-feeding are met the woman is using LAM. In this sense it is very different from the other methods (but you are allowing for this. You could comment in the manuscript). 

RESPONSE: We have added information into the text that we only coded LAM use if the woman reported that this was the method she was using to avoid a pregnancy and not based on the conditions listed above. 

• Also, you cannot be using LAM for more than 6 months. It is therefore not a solution as a post-partum method (just for the immediate postpartum). I would like to know how this is captured in the model, particularly in connection with questions 2 and 3. 

RESPONSE: As mentioned above, LAM was only coded as the postpartum method if it was the only method listed in the one-year postpartum period; any woman who reported LAM and then another method after LAM that is more effective, would be coded based on the more effective method. Only 54 women who had a birth reported only LAM as their method in the postpartum period and all of this use started immediately postpartum.

---

## [Editor Report · Decision Letter 2]

9 Dec 2021

The Direct and Indirect Effects of Community Beliefs and Attitudes on Postpartum Contraceptive Method Choice among Young Women Ages 15-24 in Nigeria

PONE-D-21-18360R2

Dear Dr. Speizer,

We’re pleased to inform you that your manuscript has been judged scientifically suitable for publication and will be formally accepted for publication once it meets all outstanding technical requirements.

Kind regards,

José Antonio Ortega, Ph.D.

Academic Editor

PLOS ONE

Additional Editor Comments (optional):

The manuscript has improved much in terms of reproducibility and completeness of the methods section, something particularly important in this contribution. It now fulfills PLOS ONE criteria. The descriptions added regarding the software in the main text and appendices 1 and 2 are also appropriate.

Please only correct some typos before publication:

- In the main text, footnote 3 please write "The Fortran program" instead of the "The fortran program". Please also indicate the version used. In the appendix you mention Fortran 77. If it is the case, just mention The Fortran 77 program ... (otherwise specify).

In appendix S2 there are two typos:

- The equation in page 1 does not read properly. I assume that the first PROD SUM combination is shown as 11 E. Please correct.

- Just after the equation, please remove the space in "observa tions".
---

## [Editor Report · Acceptance letter]

6 Jan 2022

PONE-D-21-18360R2 

The direct and indirect effects of community beliefs and attitudes on postpartum contraceptive method choice among young women ages 15-24 in Nigeria 

Dear Dr. Speizer:

I'm pleased to inform you that your manuscript has been deemed suitable for publication in PLOS ONE. Congratulations! Your manuscript is now with our production department. 

Kind regards, 

on behalf of

Dr. José Antonio Ortega 

Academic Editor

PLOS ONE